# Relationship Between Lower-Extremity Co-Contraction and Jerk During Gait

**DOI:** 10.3390/s25072327

**Published:** 2025-04-06

**Authors:** Toshinori Miyashita, Kengo Kawanishi, Shintarou Kudo

**Affiliations:** 1Inclusive Medical Sciences Research Institute, Morinomiya University of Medical Sciences, Nankokita 1–26–16, Suminoe Ward, Osaka 559-8611, Japan; toshinori_miyashita@morinomiya-u.ac.jp (T.M.); kengo_kawanishi@morinomiya-u.ac.jp (K.K.); 2Osaka Bay Clinic, Morinomiya University of Medical Sciences, Nankokita 1–26–16, Suminoe Ward, Osaka 559-8611, Japan; 3Department of Physical Therapy, Morinomiya University of Medical Sciences, Nankokita 1–26–16, Suminoe Ward, Osaka 559-8611, Japan; 4AR-Ex Medical Research Center, 4–13–1, Todoroki Setagaya Ward, Tokyo 158-0082, Japan

**Keywords:** elderly, IMU, gait, co-contraction

## Abstract

**Highlights:**

**What are the main findings?**
Our study estimated the lower-extremity co-contractions from lower-leg jerks during gait.Our study showed that multiple regression analyses adjusted for age and gait speed revealed a relationship between jerks and co-contraction index.

**What is the implication of the main finding?**
The study showed that reducing co-contraction reduces joint load, which is important for increasing lifespan.The gait analysis developed in this study can be a meaningful method for early preventive interventions for decreased gait speed.

**Abstract:**

The elderly exhibit increased co-contraction (CC) during gait, reducing movement smoothness. The jerk has been used to quantitatively smoothness. This study aimed to investigate the relationship between lower-leg jerk and lower-extremity CC during gait. Participants were 30 healthy middle-aged and elderly people. Surface electromyography (EMG) was measured from the tibialis anterior (TA), gastrocnemius lateralis (GL), vastus lateralis (VL), and biceps femoris (BF). An inertial measurement unit was attached to the lower-leg. Jerk was calculated from inertial measurement unit (IMU) acceleration data, and CC was quantified as the percent co-contraction index (CCI) for TAGL, VLBF, and VLGL. To examine the correlation between CCI and jerk, the part with the highest correlation between jerk and CC during gait was used as the dependent variable, and a multiple regression analysis was performed to obtain the estimated CC values (*p* < 0.05). VLGL CCI increased with higher jerk during the second half of the stance phase and also increased as gait speed declined. The CCI of the VLGL in-creased with age. The multiple regression analysis adjusted for age and gait speed revealed a relationship between jerks and CCI. The CCI of the VLGL is most closely related to lower-leg jerks, which affect the gait of the elderly.

## 1. Introduction

In Japan, “Locomotive syndrome” (LS) refers to a condition characterized by reduced mobility, such as difficulty in sit-to-stand transitions or gait, resulting from impairment of locomotive organs. LS is considered a problem to prevent [1] because of its close relationship with walking disorders, and maintaining and improving walking ability is an important social need. The decline in gait speed that occurs with aging [2,3] causes many adverse events [2,4,5,6] therefore, prevention is necessary.

Factors that contribute to decreased gait speed in the elderly include decreased hip extension and ankle plantar flexion angles in the late stance phase, decreased ankle plantar flexor power (AP) [7,8,9,10,11], and increased lower-extremity CC [12]. To prevent a decline in walking ability in the elderly, evaluating performance, such as walking speed, and analyzing the factors that influence the decline, are necessary.

Gait analysis of the elderly in clinical practice often involves observational gait analysis, gait speed, and stride time variability (STV), a measure of stability during gait, with an accelerometer [5,13].

These can provide an understanding of walking ability or gait patterns, however, they cannot analyze the factors causing a decrease in gait speed. Recently, analyzing joint angles and estimating ankle joint power during gait in clinical practice by simply using an IMU on the body has become possible [14,15,16,17,18,19], however, IMU-based methods have not been developed for gait analysis of CC, which is a factor in decreasing gait speed in the elderly.

The elderly increase their CC during gait to increase joint stiffness and stability during walking [20]. However, a higher CC results in increased energy cost [21], risk of falling [22], and decreased gait speed [12]. Specifically, increased CC causes adverse events owing to the loss of smooth movement in exchange for stability during gait. The jerk, a derivative of acceleration, has been used to quantitatively smooth motion during movement [23,24,25]. A higher jerk is a state with a larger change in force, which means a decrease in the coordination and smoothness of movement [23,26]. Therefore, we hypothesized that CC is related to the smoothness, such that higher CC correlates with increased jerk during gait in healthy middle-aged and elderly individuals, focusing on jerks during gait and considering that a relationship exists between jerks and CC. Although previous studies have examined the effects of aging on arm swing using jerk during gait in healthy individuals [27], no studies have analyzed the gait of healthy middle-aged or elderly individuals using lower-leg jerks.

This study aimed to investigate the relationship between lower-leg jerk and lower-extremity CC during gait in healthy middle-aged and elderly individuals.

## 2. Materials and Methods

### 2.1. Study Design and Setting

This study is descriptive and was conducted at an orthopedic clinic between May 2023 and May 2024. This study was approved by the Morinomiya University of Medical Sciences Ethics Committee (approval number: 2022-006), and all participants provided written informed consent [Standard reporting guidelines].

### 2.2. Participants

Participants in this study included 32 middle-aged and elderly people (65.4 ± 12.1 years, 160.8 ± 10.2 cm, 62.9 ± 12.3 kg) in the community-dwelling. Participants were admitted to the study if they fulfilled the following inclusion criteria: (1) had never used long-term care insurance services or nursing care services, (2) were middle-aged or older, (3) were able to walk independently, and (4) had a stable general physical condition that would allow them. Participants were excluded if they (1) had any condition that would affect their gait performance, (2) had severe lower extremity limited joint range of motion, (3) had pain during walking, (4) had an unstable physical condition, or (5) needed walking aids or any manual assistance.

### 2.3. Outcome Measures

The outcomes were the CC values (CCI) of the three lower extremity parts (shank, thigh, and thigh-shank), the difference value as the maximum and minimum changes in lower-leg jerks during the stance phase, estimated AP, STV, and comfortable gait speed. Each participant was instructed to walk at a comfortable speed.

#### 2.3.1. Surface EMG

Surface EMG signals were recorded on the TA, GL, VL, and BF using a 4-channel using the Wave COMETA EMG system (Cometa slr, Milan, Italy, 2000 Hz). EMG signals were collected at a sampling frequency of 2000 Hz, using Ag/AgCl disc electrodes (Ambu® Blue Sensor, Copenhagen, Denmark) with an active area of 1 cm2 and inter-electrode distance of 2 cm ranged in a bipolar configuration. The electrodes were positioned on the participants according to the Surface EMG for the Noninvasive Assessment of Muscles guidelines (Figure 1). Before placing the electrodes, the skin of the participants was treated appropriately to reduce impedance. The EMG signals were processed using a Butterworth bandpass fourth-order filter of 20–400 Hz, full-wave rectification, and a Butterworth low-pass fourth-order filter of 6 Hz [28]. After waveform processing, the integral value of each muscle activity was calculated.

##### Percent CCI

The CCI calculated formula in accordance with previous studies [28] (1).(1)$$PercentCo-contractionIndex(CCI)=2\times \frac{commonareaA\&B}{areaA+areaB}\times 100$$

The percent CCI was calculated as the percentage of CC between the agonist/an tangoist muscles in the shank part (TAGL), thigh part (VLBF), and thigh-shank parts (VLGL). Each CCI was obtained and defined as the entire stance phase (gait cycle: 0–62%), the first half of the stance phase (gait cycle: 0–31%), and the second half of the stance phase (gait cycle: 32–62%). Each CCI was averaged over five trials acquired per participant (Figure 2).

#### 2.3.2. IMU Settings

An IMU (Cometa slr, Milan, Italy, 143 Hz) was attached to the fibular head of the lower leg to measure the acceleration waveform data (three axes defined as Ax, vertical; Ay, anterior–posterior; and Az, mediolateral axes). The IMU was mounted and fixed to the fibular head of the lower leg using double-sided tape (Figure 1). The IMU, EMG, and videos were synchronized so that the waveform data and gait cycle could be identified. The EMG sensor with IMU were connected to an EMG data collection system with the wireless each signals and video synchronization were recorded and collected using software named EMG and Motion Tools, software version 8.7.6.0 (Cometa slr, Milan, Italy). Video synchronization settings were as follows: frame rate of 30 fps, externally recorded video (MP4 format), and real-time recording with automatic saving.

##### Jerk

The jerk follows the previous study procedure. Raw linear acceleration data in three axes (Ax, vertical; Ay, anterior–posterior; and Az, mediolateral) were low-pass Butterworth filtered at 2 Hz [24]. The jerk was calculated from the first-time derivative of linear acceleration. The time-differentiated three-axis accelerations were calculated as jerks x, y, and z. The stance phase of the gait cycle was divided into the first and second halves (Figure 3). The difference between the minimum and maximum values of each three-axis jerk was calculated for each stance phase, and the average value for five gait cycles was used. The jerk was adjusted by dividing it by the gait speed [29].

##### Estimated AP

In our previous study [19] using an IMU we developed a method to calculate the estimated AP from the vertical acceleration waveform during a walking terminal stance using an IMU. The estimated AP was calculated using the acceleration from an IMU mounted on the fibular head while the patient was walking from heel-off to toe-off at her own comfortable pace. The estimated AP is calculated as follows:(2)$$Estimated\;AP\;\left(W\right)=-4.689+(0.269\times Ax)+(0.104\times Body\;Weight)$$

Additionally, the estimated AP was normalized by body mass.

##### STV

STV was calculated as the standard deviation of each stride time divided by the mean stride time during a steady gait over five steps.

#### 2.3.3. Gait Speed

Gait speed measurements were performed on a flat walkway with a total length of 9 m, with tape marked at both ends and 2 m acceleration and deceleration distances. Each participant was instructed to walk at a comfortable speed. The comfortable walking speed, measured using a stopwatch, was averaged over the values measured twice. All the participants used everyday footwear to test their gait performance.

### 2.4. Sample Size

The sample size was calculated based on the results of a previous study [30]. As a result, the sample size was estimated for a power of 0.8 and an effect size of 0.43 in this study. The sample size was determined using G*Power 3 software (version 3.1.9.4) with a *t*-test for correlation. Twenty-nine participants were deemed adequate.

### 2.5. Statistical Analysis

The normality of the data was determined using the Kolmogorov–Smirnov test. The Pearson product-moment correlation coefficient and Spearman’s rank correlation coefficient were used to examine the correlation between each part (TAGL, VLBF, and VLGL), CCI (entire stance phase, first half of the stance phase, and second half of the stance phase), and three-axis (Ax, Ay, and Az) jerk (first half of the stance phase and second half of the stance phase). In addition, Spearman’s rank correlation coefficient was used to analyze the relationship between the integral value of each muscle activity (TA, GL, VL, BF) and each part’s CCI. The part with the highest correlation between jerk and CC during gait was used as the dependent variable, and a multiple regression analysis was performed using a stepwise method to obtain the estimated CC values. All statistical analyses were performed using IBM Statistical Package for the Social Sciences (SPSS) Statistics for Windows (IBM SPSS Statistics for Windows, Version 24.0. Armonk, NY, USA). Statistical significance was set at *p* < 0.05.

## 3. Results

This section may be divided by subheadings. It should provide a concise and precise description of the experimental results, their interpretation, as well as the experimental conclusions that can be drawn.

### 3.1. General Characteristics and Outcomes

One recruit was in an unstable physical condition, and one recruit with missing data during gait was excluded. Thirty participants completed all the assessments in this study. The general characteristics, including comfortable gait speed, estimated AP, normalized estimated AP of the participants, and CCI, are summarized in Table 1.

### 3.2. Correlation Between CCI, Age, Gait Speed and the Integral Value of Muscle Activities

A significant negative correlation was observed between jerk × (second half of the stance phase) and VLGL (entire stance phase (r = 0.59, *p* < 0.01), first half of the stance phase (r = 0.44, *p* < 0.05), and second half of the stance phase (r = 0.39, *p* < 0.05)). Additionally, significant positive correlations were observed between jerk × (second half of the stance phase) and TAGL (second half of the stance phase (r = 0.38, *p* < 0.05)), and a significant positive correlation between the lateral direction as jerk z (second half of the stance phase) and VLBF (second half of the stance phase (r = 0.38, *p* < 0.05)) (Table 2). A significantly negative correlation was observed between VLGL (entire stance phase, second half of stance phase) and gait speed (entire stance phase; r = −0.36, *p* < 0.05; second half of stance phase; r = −0.53, *p* < 0.01). A significant positive correlation was ob-served between the VLGL (the second half of the stance phase) and age (r = 0.50, *p* < 0.01). A significant positive correlation was observed between the gait speed and estimated AP (r = 0.38, *p* < 0.05). No significant correlations were observed (Table 2). There was a significant negative correlation between GL muscle activity during the second half of the stance phase and the CCI of VLGL (r = −0.41, *p* < 0.05) and VLBF (r = −0.43, *p* < 0.05) during the same phase (Table 3). Figure 4 shows the five age strata of the representative cases. The GL activity decreased with age, and the VLGL (the second half of the stance phase) CCI increased with age (Figure 4).

### 3.3. Multiple Regression Analysis

The results of the stepwise multiple regression analysis are presented in Table 4. Multiple regression analysis was performed using a stepwise method to obtain the es-timated VLGL (second half of the stance phase) and CCI. The analysis revealed that the normalized jerk × (second half of the stance phase), age, and gait speed were highly significant partial regression coefficients (adjusted R^2^ = 0.53, Table 4). The results were obtained using the following equation for the estimated second half of the stance VLGL CCI:(3)$$\begin{array}{c} Estimated\;Second\;half\;of\;stance\;VLGL\;CCI\;\left(W\right)\\ \;\;\;\;\;=39.510+\left(27.105\times jerk\;x\;\left(second\;of\;half\;stance\;phase\right)\right)+\left(0.343\times Age\right)\\ \;\;\;\;\;+\left(-17.511\times Gait\;Speed\right) \end{array}$$

## 4. Discussion

This study aimed to estimate lower-extremity CC from lower-leg jerks during gait. Lower leg jerk was the relationship between CCI of the VLGL; CCI increased with in-creasing jerk, and CCI of the VLGL (entire stance phase, second half of stance phase) increased with a decline in gait speed. Additionally, the CCI of the VLGL (second half of the stance phase) increased with age. Therefore, multiple regression analysis adjusted for age and gait speed revealed a relationship between jerks and CCI. In other words, the CCI of the VLGL is most closely related to lower-leg jerks, which affect the gait of the elderly and could be a novel index for gait analysis in the elderly. Additionally, the CCI of the VLGL can be easily estimated in clinical practice by measuring the lower leg jerk based on the results of this study.

A positive correlation was observed between jerk × (second half of the stance phase) and the CCI of the VLGL (entire stance phase, first half of the stance phase, and second half of the stance phase). The jerk was used as an index to analyze changes in smoothness and force and has been applied in movement analysis studies [24,25,31]. Movement analysis using jerks for musculoskeletal disorders [24,26] and intractable neurological diseases were reported [32,33], however, the gait of healthy middle-aged and elderly individuals using lower-leg jerks. No studies have analyzed the gait of healthy middle-aged or elderly individuals using lower-leg jerks. In other words, this study is the first to estimate muscle activity from lower leg acceleration in middle-aged and elderly participants.

It has been reported that a characteristic of gait in the elderly is decreased ankle plantar flexor activity [7,8,9,10,11]. Brunner et al. reported that the ankle plantar flexors control tibial advancement and contribute to knee extension [34]. In the present study, a negative correlation was observed between GL muscle activity during the second half of the stance phase and the CCI of VLGL and VLBF during the same phase. This is considered to reflect a compensatory mechanism, in which eccentric contraction of the ankle plantar flexors attempts to control anterior tibial tilt in the second half of the stance phase. Additionally, as the downward acceleration of the tibia increases, jerk in the x-direction also increases, indicating excessive anterior tibial tilt during the second half of the stance phase. The knee joint of healthy elderly asymptomatic is not extended and is in increasing knee flexion during the late stance phase compared to younger asymptomatic [35]. In other words, the increase in jerk x may be attributed to greater control of anterior tibial tilt caused by excessive lower-extremity CC.

The results of this study indicate that the lower-extremity CCI can be analyzed using only a single IMU without EMG. In particular, the functional significance of VLGL during gait has been reported to be a “plantar flexion–knee extension couple” that facilitates smooth knee extension movement in the second half of the stance phase by knee extensors in the early stance phase and ankle plantar flexors in the late stance phase [36]. Furthermore, the results of this study showed that the CCI of the VLGL (second half of the stance phase) is related to gait characteristics with age, and that could be easily quantified VLGL can be applied to gait analysis in the elderly population. Additionally, CC is related to joint contact forces [37]. Reducing CC leads to reduced joint load and is important for increasing lifespan. Therefore, the gait analysis developed in this study can be a meaningful method for early preventive interventions for decreased gait speed.

This study had some limitations. Previous studies have reported that muscle weakness and limited range of motion can cause abnormal muscle activity [38]. The effects on the results are unknown because of the lower extremity muscle strength, joint angles during gait, and muscle activity of the vastus medialis and medial ham-strings on the medial side of the knee joint. Additionally, numerous female participants were observed; therefore, differences in muscle activity may occur due to sex differences [39]. Previous studies have also suggested that another factor that may be affected by lower-extremity CC is the influence of proprioception with aging [40] and investigating its relationship with CC in future studies is necessary. Additionally, in this method, the fibular head provides high reproducibility when palpating bony landmarks, and we consider that measurements can be easily performed even without specialized knowledge. However, since calculating jerk is complex, it will be necessary to develop software capable of automatically computing the CCI in the future.

## 5. Conclusions

The CCI increased with increasing jerk, and the CCI of the VLGL (entire stance phase and second half of the stance phase) increased with a decline in gait speed. Additionally, the CCI of the VLGL (second half of the stance phase) increased with age. Among all regions, the CCI of the VLGL was most closely related to lower-leg jerks, which influence gait in older adults, and may serve as a novel index for gait analysis in this population. Mounting a single IMU on the lower leg suggested that the acceleration waveform during gait is related to the CC of the lower leg.

## Figures and Tables

**Figure 1 sensors-25-02327-f001:**
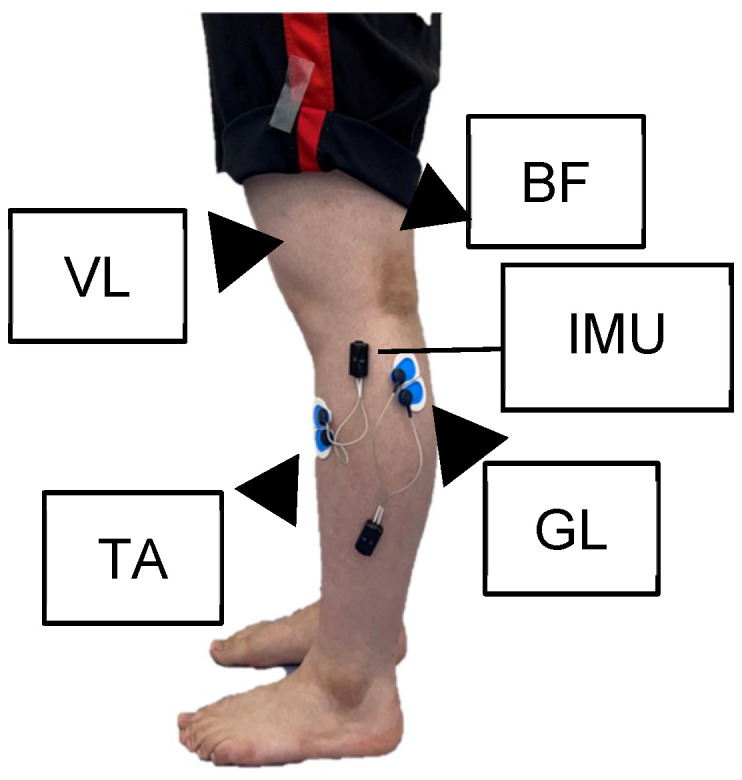
Electromyogram electrode position and IMU location. Surface EMG signal were recorded on the tibialis anterior (TA), gastrocnemius lateralis (GL), vastus lateralis (VL), and biceps femoris (BF) using a 4-channel using the Wave COMETA EMG system (2000 Hz. Milan, Italy). The electrodes were positioned on the participants’ according to the EMG for the Non-invasive Assessment of Muscles guidelines. An IMU (Cometa slr, Milano, Italia, 143 Hz) was attached to the fibular head of the lower leg to measure acceleration waveform data (three axes defined as; Ax: vertical, Ay: anterior–posterior, Az: mediolateral axes). The IMU was mounted on and fixed to the fibular head of the lower leg using double-sided tap.

**Figure 2 sensors-25-02327-f002:**
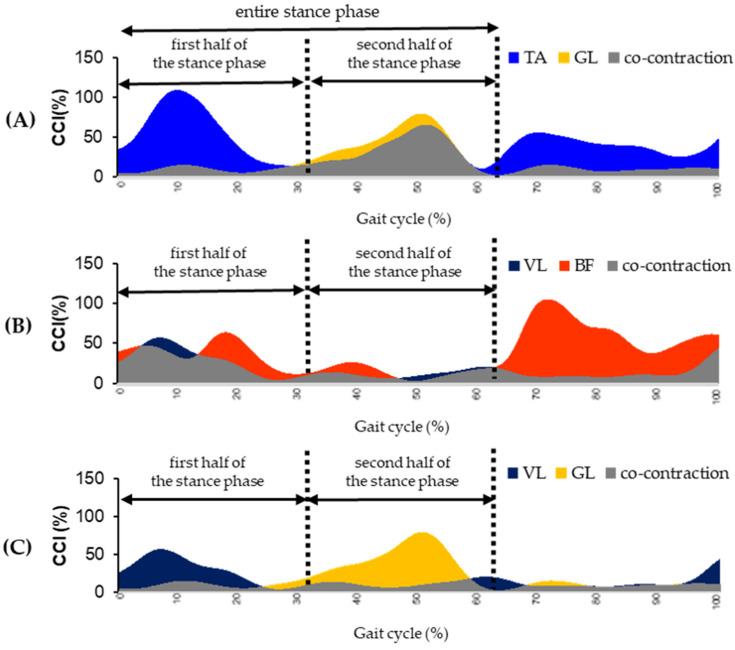
Lower-extremity muscle activity and CCI during gait cycle. (**A**) shank part (TAGL), (**B**) thigh part (VLBF), (**C**) thigh-shank part (VLGL). The percent CCI was calculated as the percentage of CC between the agonist/an tangoist muscles in the shank part (TAGL), thigh part (VLBF), and thigh-shank parts (VLGL). Each CCI was obtained and defined as the entire stance phase (gait cycle: 0–62%), the first half of the stance phase (gait cycle: 0–31%), and the second half of the stance phase (gait cycle: 32–62%). Each CCI was averaged over five trials acquired per participant.

**Figure 3 sensors-25-02327-f003:**
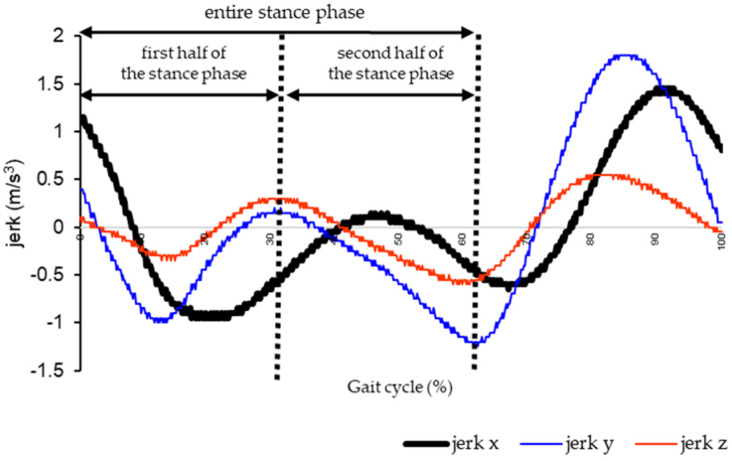
**Lower-leg jerk during gait cycle.** The stance phase of the gait cycle was divided into the first half of the stance phase and the second half of the stance phase. The difference value between the minimum and maximum values of each three axes jerk were calculated for each stance phase.

**Figure 4 sensors-25-02327-f004:**
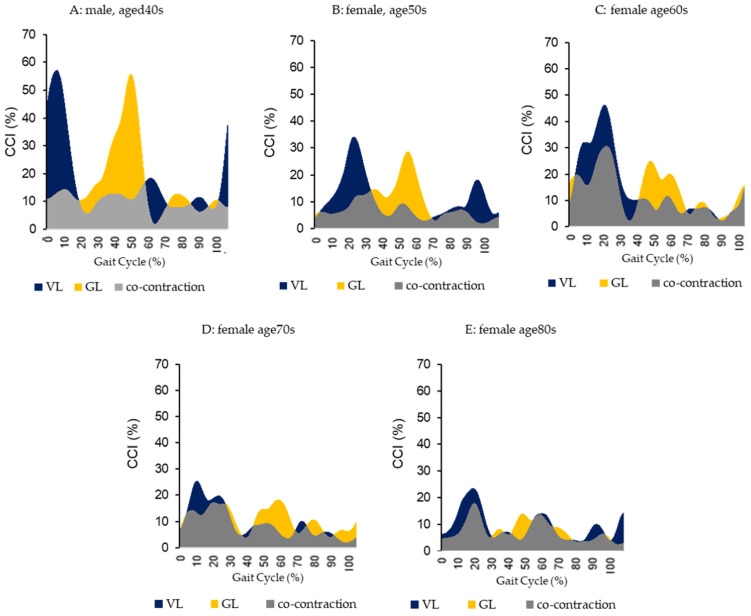
CCI of VLGL in different age strata. This figure was presented among the five different age strata of cases. GL activity decreases with age, VLGL (the second half of stance phase) CCI relatively increases with age (Figure 4). Gait speed, VL/GL (second half of stance phase). (gait speed, CCI of VLGL; (**A**): 1.46 m/s, 41.6% (**B**): 1.43 m/s, 44.1% (**C**): 1.55 m/s, 65.1% (**D**): 1.12 m/s, 72.7% (**E**): 0.93 m/s, 81.3%).

**Table 1 sensors-25-02327-t001:** Summary of general characteristics and outcomes.

Variable	*n* = 30 (Female: 22, Male: 8)
Age (year)	66.0 (12.0)
Height (cm)	159.9 (9.8)
Weight (kg)	61.5 (10.8)
BMI	24.0 (3.7)
Gait speed (m/s)	1.22 (0.22)
Estimated AP (W/kg)	0.065 (0.010)
STV (%)	2.18 (0.92)
TAGL (%)	62.2 (11.6)
VLBF (%)	64.4 (10.1)
VLGL (%)	59.8 (12.3)

Average (standard deviation), Body mass index (BMI), ankle plantar flexor power (AP), stride time variability (STV), tibialis anterior (TA), gastrocnemius lateralis (GL), vastus lateralis (VL), biceps femoris (BF).

**Table 2 sensors-25-02327-t002:** Correlation between lower-leg jerks (jerk x, jerk y, jerk z) and CCI, age, and gait speed.

			First Half of Stance Phase			Second Half of Stance Phase			Variables		
			Jerk x	Jerk y	Jerk z	Jerk x	Jerk y	Jerk z	AP	GS	Age
CCI	Entire stance phase	TAGL	0.17	0.02	−0.002	0.24	0.003	−0.08	−0.009	−0.23	0.11
		VLBF	0.05	−0.32	−0.27	0.18	−0.28	−0.09	−0.007	−0.04	0.13
		VLGL	0.30	0.003	0.15	**0.59 *****	−0.20	0.02	−0.004	**−0.36 ***	0.27
	First half of stance phase	TAGL	0.31	0.13	0.05	0.27	−0.02	−0.06	0.01	−0.09	0.11
		VLBF	0.05	−0.22	−0.12	0.12	−0.16	0.06	0.05	−0.09	0.13
		VLGL	0.23	0.02	0.26	**0.44 ***	0.01	0.20	0.15	−0.11	0.10
	Second half of stance phase	TAGL	0.19	−0.02	−0.19	0.38 *	−0.17	−0.26	−0.06	−0.30	0.24
		VLBF	0.21	−0.32	**−0.38 ***	0.17	−0.27	−0.20	0.05	0.10	0.06
		VLGL	0.23	0.05	−0.03	**0.39 ***	−0.31	−0.25	−0.18	**−0.53 ****	**0.50 ****

* *p* < 0.05, ** *p* < 0.01, *** *p*<0.001

**Table 3 sensors-25-02327-t003:** Correlation between the integral value of muscle activities and each part’s CCI.

			Entire Stance Phase			First Half of Stance Phase			Second Half of Stance Phase		
			TAGL	VLBF	VLGL	TAGL	VLBF	VLGL	TAGL	VLBF	VLGL
The integral value of each muscle activity	Entire stance phase	TA	−0.14	−0.1	−0.01	−0.15	−0.31	0.13	0.07	−0.02	0.01
		GL	−0.15	**−0.50 ****	−0.22	0.08	**−0.47 ****	0.13	−0.33	−0.35	−0.31
		VL	0.02	−0.21	**−0.42 ***	−0.01	**−0.39 ***	−0.53	−0.11	−0.21	0.01
		BF	−0.05	−0.28	**−0.42 ***	−0.05	−0.32	**−0.40 ****	−0.1	−0.22	**−0.42 ***
	First half of stance phase	TA	−0.21	−0.19	0.03	−0.22	−0.35	0.15	0.01	−0.11	0.10
		GL	0.15	−0.18	0.15	0.35	−0.29	0.09	−0.08	−0.16	0.14
		VL	−0.05	−0.24	**−0.44 ***	−0.04	**−0.45 ***	**−0.55 ****	−0.16	−0.21	−0.05
		BF	−0.07	−0.06	−0.33	−0.09	−0.27	**−0.39 ****	−0.03	−0.02	−0.33
	Second half of stance phase	TA	0.09	−0.07	−0.16	0.02	−0.21	0.15	0.23	0.02	0.02
		GL	−0.3	**−0.56 ****	−0.25	−0.12	**−0.39 ***	−0.15	**−0.38 ***	**−0.43 ***	**−0.41 ***
		VL	0.02	−0.14	**−0.42 ***	0.03	−0.25	**−0.48 ****	−0.11	−0.19	−0.11
		BF	−0.05	**−0.40 ***	**−0.44 ***	−0.03	−0.32	**−0.37 ***	−0.15	**−0.37 ***	−0.44

* *p* < 0.05, ** *p* < 0.01.

**Table 4 sensors-25-02327-t004:** Multiple regression analysis.

	Unstandardized Coefficients	Standardized Coefficients β	95% Confidence Interval		*p*-Value	VIF
			Lower Bound	Upper Bound		
(Constant)	39.510		7.245	71.775	0.018	
Second half of stance Ax jerk	27.105	0.393	8.049	46.161	0.007	1.116
Age	0.343	0.348	0.072	0.613	0.015	1.106
gait speed	−17.511	−0.328	−32.423	−2.599	0.023	1.142

Adjusted R^2^ = 0.53.

## Data Availability

The data presented in this study are available on request from the corresponding author.

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
