# Peer review of "Relationship Between Lower-Extremity Co-Contraction and Jerk During Gait"

_sensors, 2025, doi:10.3390/s25072327_

Round 1

Reviewer 1 Report

Comments and Suggestions for Authors

General Comments

The manuscript presents an interesting and novel investigation into the relationship between lower-leg jerk and lower-extremity co-contraction during gait in healthy middle-aged and elderly individuals. The use of inertial measurement units to estimate co-contraction without relying on electromyography (EMG) is a promising approach for clinical gait analysis, particularly for aging populations. The study addresses a relevant issue in the context of "Locomotive Syndrome" and declining gait speed, which are significant public health concerns. The methodology is generally sound, and the statistical analysis is appropriate for the research question. However, the manuscript would benefit from improved clarity, grammatical corrections, and a more thorough discussion of the findings' implications and limitations. Additionally, some sections lack sufficient detail or justification, and the presentation of results could be enhanced for better readability and interpretation.

Specific Comments

Abstract

  1. Lines 23-24: "The elderly increase their co-contraction (CC) during gait and the loss of smooth movement" is awkwardly phrased. Consider revising it to "The elderly exhibit increased co-contraction (CC) during gait, resulting in reduced movement smoothness."
  2. Line 29: The abstract mentions surface EMG and IMU but lacks detail on how jerk and CC were measured or correlated. Add a brief description, e.g., "Jerk was derived from IMU acceleration data, and CC was calculated as a percent co-contraction index (CCI)."
  3. Line 34: "The CCI increased with increasing jerk" is vague. Specify which CCI (e.g., VLGL) and under what conditions (e.g., stance phase) to strengthen the summary.

Introduction

  1. Line 64: The term "Locomotive Syndrome" is introduced without sufficient explanation for an international audience. Define it more clearly. e.g., "Locomotive Syndrome (LS), a condition prevalent in Japan, refers to impaired mobility due to musculoskeletal
  2. Line 80: The claim "No studies have analyzed the gait of healthy middle-aged or elderly individuals using lower-leg jerks" needs substantiation.
  3. Lines 66-67: "We hypothesized that CC is related to the smoothness of gait" is vague. Specify the expected direction (e.g., "higher CC correlates with increased jerk") for a testable hypothesis.

Methods

  1. Line 125: The IMU was attached to the fibular head, but no rationale is provided. Explain why this location was chosen over others (e.g., shank or foot).
  2. Lines 108-109: The Butterworth filter settings (20-400 Hz bandpass, 6 Hz low-pass) are listed but not justified. Provide a brief rationale or reference supporting these choices.
  3. Line 129: The authors mention synchronizing EMG and IMU with video but do not specify the video type, sampling frequency, or synchronization method. Provide these details for reproducibility.

4.line135: A 2 Hz low-pass Butterworth filter is applied to raw acceleration data, but 2 Hz seems low and may filter out high-frequency signals. Justify this cutoff or cite evidence supporting its use.

  1. Line 140: "The jerk was adjusted by dividing it by the gait speed" is unclear. Explain why this adjustment was necessary and how it affects interpretation.

Results

  1. Line 237: Table 1 presents CCI values for TAGL, VLBF, and VLGL without defining these abbreviations in the table or addressing the female bias (22 females vs. 8 males). Define terms and justify or discuss the sex imbalance’s implications for generalizability.
  2. Figure 4 Description (Line 262): The figure caption mentions five age strata but lacks detail on how cases were selected as "representative." Provide selection criteria or remove "representative" if arbitrary.

Discussion

  1. Lines 275-276: "The CCI of the VLGL can be easily estimated in clinical practice" assumes accessibility of IMUs and expertise, which may not be universal. Qualify this claim with conditions or limitations.
  2. Line 289: The discussion of ankle plantar flexor weakness and knee is plausible but lacks direct evidence from this study. Link it to specific results (e.g., EMG data).

Conclusion

  1. Line 320-321: The conclusion is brief, omitting key findings (e.g., VLGL CCI’s correlation with jerk, age, and gait speed) and future research directions.

Comments on the Quality of English Language

The English could be improved to more clearly express the research.

Author Response

Specific Comments

Abstract

Comments 1: Lines 23-24: "The elderly increase their co-contraction (CC) during gait and the loss of smooth movement" is awkwardly phrased. Consider revising it to "The elderly exhibit increased co-contraction (CC) during gait, resulting in reduced movement smoothness."

Response 1: As you suggested, we have corrected the sentence (Page 1, lines 23-24) as follows:

“The elderly exhibit increased co-contraction (CC) during gait, reducing movement smoothness.”

Comments 2: Line 29: The abstract mentions surface EMG and IMU but lacks detail on how jerk and CC were measured or correlated. Add a brief description, e.g., "Jerk was derived from IMU acceleration data, and CC was calculated as a percent co-contraction index (CCI)."

Response 2: As you commented, we have corrected the sentence (Page 1, lines 29-30) as follows:

“Jerk was calculated from IMU acceleration data, and CC was quantified as the percent co-contraction index (CCI) for TAGL, VLBF, and VLGL.”

Comments 3: Line 34: "The CCI increased with increasing jerk" is vague. Specify which CCI (e.g., VLGL) and under what conditions (e.g., stance phase) to strengthen the summary.

Response 3: As you suggested, we have corrected the sentence (Page 1, lines 33-34) as follows:

“VLGL CCI increased with higher jerk during the second half of the stance phase and also increased as gait speed declined.”

Introduction

Comments 4: Line 64: The term "Locomotive Syndrome" is introduced without sufficient explanation for an international audience. Define it more clearly. e.g., "Locomotive Syndrome (LS), a condition prevalent in Japan, refers to impaired mobility due to musculoskeletal

Response 4: As you commented, we have corrected the sentence and defined LS (Page 2, lines 42-44) as follows:

“In Japan, “Locomotive syndrome” (LS) refers to a condition characterized by reduced mobility, such as difficulty in sit-to-stand transitions or gait, resulting from impairment of loco-motive organs. LS is considered a problem to prevent [1]”

Comments 5: Line 80: The claim "No studies have analyzed the gait of healthy middle-aged or elderly individuals using lower-leg jerks" needs substantiation.

Response 5: Thank you for your constructive feedback. We have corrected the sentence and added the appropriate reference (Page 2, lines 71-73) as follows:

“Although previous studies have examined the effects of aging on arm swing using jerk during gait in healthy individuals [27], no studies have analyzed the gait of healthy middle-aged or elderly individuals using lower-leg jerks.”

Added reference:

  1. Mirelman, A.; Bernad-Elazari, H.; Nobel, T.; Thaler, A.; Peruzzi, A.; Plotnik, M.; Giladi, N.; Hausdorff, J.M. Effects of Aging on Arm Swing during Gait: The Role of Gait Speed and Dual Tasking. PLOS One 2015, 10, e0136043, DOI:10.1371/journal.pone.0136043.

Comments 6: Lines 66-67: "We hypothesized that CC is related to the smoothness of gait" is vague. Specify the expected direction (e.g., "higher CC correlates with increased jerk") for a testable hypothesis.

Response 6: As suggested, we have corrected the sentence (Page 2, lines 69) as follows:

“Therefore, we hypothesized that CC is related to the smoothness, such that higher CC correlates with increased jerk during gait in healthy middle-aged and elderly individuals,”

Methods

Comments 7: Line 125: The IMU was attached to the fibular head, but no rationale is provided. Explain why this location was chosen over others (e.g., shank or foot).

Response 7: This response has been added to the study limitations in relation to Comment 14.

The fibular head provides high reproducibility when palpating bony landmarks; therefore, we considered that measurements can be easily performed even without specialized knowledge.

Comments 8: Lines 108-109: The Butterworth filter settings (20-400 Hz bandpass, 6 Hz low-pass) are listed but not justified. Provide a brief rationale or reference supporting these choices.

Response 8: Thank you for your meaningful comment. The filter settings were directly taken from citation number 28. The participants’ age range was similar, and the filter settings followed the procedure described in the previous study by J. Lo et al. [28].

Comments 9: Line 129: The authors mention synchronizing EMG and IMU with video but do not specify the video type, sampling frequency, or synchronization method. Provide these details for reproducibility.

Response 9: We have added the video settings (Page 3, lines 133-135) as follows:

“Video synchronization settings were as follows: frame rate of 30 fps, externally recorded video (MP4 format), and real-time recording with automatic saving.”

Comments 10: 4.line135: A 2 Hz low-pass Butterworth filter is applied to raw acceleration data, but 2 Hz seems low and may filter out high-frequency signals. Justify this cutoff or cite evidence supporting its use.

Response 10: We apologize for the omission of the citation. The citation number has now been added (Page 3, lines 138-140) as follows:

“The jerk follows the previous study procedure. Raw linear acceleration data in three axes (Ax, vertical; Ay, anterior-posterior; and Az, mediolateral) were low-pass Butterworth filtered at 2 Hz [24].”

Comments 11: Line 140: "The jerk was adjusted by dividing it by the gait speed" is unclear. Explain why this adjustment was necessary and how it affects interpretation.

Response 11: In accordance with your comment, we have added a reference to the sentence.

Because acceleration is affected by gait speed, we adjusted for this by normalizing acceleration to reduce inter-participant variability.

The citation number has now been added (Page 4, lines 145-146) as follows:

“The jerk was adjusted by dividing it by the gait speed [29].”

Added reference:

  1. Craig, J.J.; Bruetsch, A.P.; Huisinga, J.M. Coordination of Trunk and Foot Acceleration during Gait Is Affected by Walking Velocity and Fall History in Elderly Adults. Aging Clin. Exp. Res. 2019, 31, 943–950, DOI:10.1007/s40520-018-1036-4.

Results

Comments 12: Line 237: Table 1 presents CCI values for TAGL, VLBF, and VLGL without defining these abbreviations in the table or addressing the female bias (22 females vs. 8 males). Define terms and justify or discuss the sex imbalance’s implications for generalizability.

Response 12: We sincerely apologize. The abbreviations in Table 1 have been revised (Page 6, Table 1).

Additionally, a reference regarding sex differences has been added to the limitations section.

Regarding sex imbalances, it has been reported that females exhibit higher muscle activity for ankle joint stabilization.

“differences in muscle activity may occur due to sex differences [39].” (Page 10, line 357)

Added reference:

  1. Mengarelli, A.; Maranesi, E.; Burattini, L.; Fioretti, S.; Di Nardo, F. Co-Contraction Activity of Ankle Muscles during Walking: A Gender Comparison. Biomed. Signal Process. Control 2017, 33, 1–9, DOI:10.1016/j.bspc.2016.11.010.

Comments 13: Figure 4 Description (Line 262): The figure caption mentions five age strata but lacks detail on how cases were selected as "representative." Provide selection criteria or remove "representative" if arbitrary.

Response 13: As you suggested, we have removed the word “representative” (Page 9, lines 304, Figure 4):

“This figure presents data across five different age strata.

GL activity decreases with age, whereas the VLGLCCI during the second half of the stance phase relatively increases with age (Figure 4).”

Discussion

Comments 14: Lines 275-276: "The CCI of the VLGL can be easily estimated in clinical practice" assumes accessibility of IMUs and expertise, which may not be universal. Qualify this claim with conditions or limitations.

Response 14: We agree with your comment. As you suggested, this response has been added to the study limitations (Page 10, lines 360-364):

“Additionally, in this method, the fibular head provides high reproducibility when palpating bony landmarks, and we consider that measurements can be easily performed even without specialized knowledge. However, since calculating jerk is complex, it will be necessary to develop software capable of automatically computing the CCI in the future.”

Comments 15: Line 289: The discussion of ankle plantar flexor weakness and knee is plausible but lacks direct evidence from this study. Link it to specific results (e.g., EMG data).

Response 15: Thank you for your valuable comment.

             Three additions and corrections have been made:

① Additional analysis was performed (Table 3) (Page 4, lines 182-184):

“In addition, Spearman’s rank correlation coefficient was used to analyze the relationship between the integral value of each muscle activity (TA, GL, VL, BF) and each part’s CCI.”

② The results of additional analysis and Table 3 have been added (Page 5, lines 202, 215-218; Page 8, Results):

“There was a significant negative correlation between GL muscle activity during the second half of the stance phase and the CCI of VLGL (r=-0.41, p<0.05) and VLBF (r=-0.43, p<0.05) during the same phase (Table 3).”

Table 3. Correlation between the integral value of muscle activities and each part’s CCI.

Entire stance phase

First half of stance phase

Second half of stance phase

TAGL

VLBF

VLGL

TAGL

VLBF

VLGL

TAGL

VLBF

VLGL

The integral value of each muscle activity

Entire stance phase

TA

-0.14

-0.1

-0.01

-0.15

-0.31

0.13

0.07

-0.02

0.01

GL

-0.15

-0.50**

-0.22

0.08

-0.47**

0.13

-0.33

-0.35

-0.31

VL

0.02

-0.21

-0.42*

-0.01

-0.39*

-0.53

-0.11

-0.21

0.01

BF

-0.05

-0.28

-0.42*

-0.05

-0.32

-0.40**

-0.1

-0.22

-0.42*

First half of stance phase

TA

-0.21

-0.19

0.03

-0.22

-0.35

0.15

0.01

-0.11

0.10

GL

0.15

-0.18

0.15

0.35

-0.29

0.09

-0.08

-0.16

0.14

VL

-0.05

-0.24

-0.44*

-0.04

-0.45*

-0.55**

-0.16

-0.21

-0.05

BF

-0.07

-0.06

-0.33

-0.09

-0.27

-0.39**

-0.03

-0.02

-0.33

Second half of stance phase

TA

0.09

-0.07

-0.16

0.02

-0.21

0.15

0.23

0.02

0.02

GL

-0.3

-0.56**

-0.25

-0.12

-0.39*

-0.15

-0.38*

-0.43*

-0.41*

VL

0.02

-0.14

-0.42*

0.03

-0.25

-0.48**

-0.11

-0.19

-0.11

BF

-0.05

-0.40*

-0.44*

-0.03

-0.32

-0.37*

-0.15

-0.37*

-0.44

*p<0.05, **p<0.01

③ The sentence has been significantly revised (Page 10, lines 328-340) as follows:

“It has been reported that a characteristic of gait in the elderly is decreased ankle plantar flexor activity [7-11]. Brunner et al. reported that the ankle plantar flexors control tibial advancement and contribute to knee extension [34]. In the present study, a negative correlation was observed between GL muscle activity during the second half of the stance phase and the CCI of VLGL and VLBF during the same phase. This is considered to reflect a compensatory mechanism, in which eccentric contraction of the ankle plantar flexors attempts to control anterior tibial tilt in the second half of the stance phase. Additionally, as the downward acceleration of the tibia increases, jerk in the x-direction also increases, indicating excessive anterior tibial tilt during the second half of the stance phase. The knee joint of healthy elderly asymptomatic is not extended and is in increasing knee flexion during the late stance phase compared to younger asymptomatic [35]. In other words, the increase in jerk x may be attributed to greater control of anterior tibial tilt caused by excessive lower-extremity CC.”

Conclusion

Comments 16: Line 320-321: The conclusion is brief, omitting key findings (e.g., VLGL CCI’s correlation with jerk, age, and gait speed) and future research directions.

Response 16: As suggested, the following sentence has been added (Page 10, lines 366-370): 

“The CCI increased with increasing jerk, and the CCI of the VLGL (entire stance phase and second half of the stance phase) increased with a decline in gait speed. Additionally, the CCI of the VLGL (second half of the stance phase) increased with age. Among all regions, the CCI of the VLGL was most closely related to lower-leg jerks, which influence gait in older adults, and may serve as a novel index for gait analysis in this population.”

Reviewer 2 Report

Comments and Suggestions for Authors

In this study, the relationship between lower extremity co-contraction (CCI) and jerk (jerk) in middle-aged and elderly people was investigated. There are some questions for the authors:

  1. Some abbreviations are not defined when they first appear (e.g. AP, STV) and need to be fully stated.
  2. Please specify the EMG filtering parameters, IMU sampling frequency rationality, electrode /IMU fixing method.
  3. Check the “low-er-”in line 71.
  4. The number of subtitle such as 2.3.2.1 is too long, please revise it.
  5. Please check the label of figures 2, 3, and 4 are gait cycle(%), but the coordinate value are 0-1, please revise them.
  6. The sex ratio of the sample was unbalanced (22 females and 8 males), and the potential effect of sex differences on the results (e.g., differences in muscle activity) needed to be discussed.
  7. The conclusion can be extended.

Comments on the Quality of English Language

Can be improved.

Author Response

Comments 1: Some abbreviations are not defined when they first appear (e.g. AP, STV) and need to be fully stated.

Response 1: Thank you for your comment. As you noted, some terms were not properly defined. We have the revised sentences accordingly (Pages 3-4 and 6; lines 96, 98, 102 ; Table 1) to clarify the following terms:

(co-contraction index [CCI]), estimated AP, tibialis anterior (TA) and Estimated AP.

Comments 2: Please specify the EMG filtering parameters, IMU sampling frequency rationality, electrode /IMU fixing method.

Response 2: Thank you for your thoughtful comment.

① EMG Filtering Parameters

The EMG filtering parameters are described in lines 110-113.

The participants’ age ranges were similar, and the filter settings followed the procedure used in a previous study [28].

② IMU settings

The default sampling rate of the IMU was 143Hz. Previous studies, including the following reference, have reported that a sampling rate of 100 Hz is valid for gait analysis using IMUs. Therefore, we considered this setting to be appropriate.

(Reference)

Chigateri, N.G.; Kerse, N.; Wheeler, L.; MacDonald, B.; Klenk, J. Validation of an Accelerometer for Measurement of Activity in Frail Older People. Gait Posture 2018, 66, 114–117, DOI:10.1016/j.gaitpost.2018.08.024.

③ IMU fixation

The IMU fixation method is described in lines 130-131. The IMU was mounted and secured to the fibular head of the lower leg using double-sided tape.

④ Electrode fixation method

This study used electrodes with high adhesive strength and strong fixation to ensure signal stability during gait.

Comments 3: Check the “low-er-”in line 71.

Response 3: We have removed the hyphen (Page 2, line 75).

Comments 4: The number of subtitle such as 2.3.2.1 is too long, please revise it.

Response 4: Thank you for your important comment. We understand that the subtitle for section 2.3.2 (Measurement of acceleration waveform data using IMU) was too long. We have revised it as follows: “2.3.2. IMU settings” (Page 3, line 127).

Comments 5: Please check the label of figures 2, 3, and 4 are gait cycle(%), but the coordinate value are 0-1, please revise them.

Response 5: We have revised Figures 2, 3, and 4.

Comments 6: The sex ratio of the sample was unbalanced (22 females and 8 males), and the potential effect of sex differences on the results (e.g., differences in muscle activity) needed to be discussed.

Response 6: A reference regarding sex differences has been added to the limitations section.

Regarding sex imbalances, it has been reported that females exhibit higher muscle activity for ankle joint stabilization.

“differences in muscle activity may occur due to sex differences [39].”

(Page 10, line 357)

Added reference:

  1. Mengarelli, A.; Maranesi, E.; Burattini, L.; Fioretti, S.; Di Nardo, F. Co-Contraction Activity of Ankle Muscles during Walking: A Gender Comparison. Biomed. Signal Process. Control 2017, 33, 1–9, DOI:10.1016/j.bspc.2016.11.010.

Comments 7: The conclusion can be extended.

Response 7: As suggested, the following sentences have been added (Page 10, lines 366-370):

“The CCI increased with increasing jerk, and the CCI of the VLGL (entire stance phase and second half of the stance phase) increased with a decline in gait speed. Additionally, the CCI of the VLGL (second half of the stance phase) increased with age. Among all regions, the CCI of the VLGL was most closely related to lower-leg jerks, which influence gait in older adults, and may serve as a novel index for gait analysis in this population.”

Round 2

Reviewer 1 Report

Comments and Suggestions for Authors

The authors have made substantial improvements to the manuscript based on the previous review. The explanations regarding data processing, filter settings, and synchronization have been clarified, and the manuscript is now more structured and coherent. However, a few issues remain:

1. lines 133-135, synchronization of EMG, IMU, and Video: The manuscript does not provide a detailed explanation of the synchronization method. It is unclear whether synchronization was achieved via hardware or software and whether external or internal synchronization was used. Clarifying this aspect would enhance the reproducibility of the study.

2. Formatting Issue (Lines 394–399): There is an overlap of text and abbreviations in this section. Please review and revise the formatting to ensure readability.

Comments on the Quality of English Language

The English could be improved to more clearly express the research.

Author Response

RESPONSE TO REVIEWERS

Thank you very much for reviewing our manuscript and offering valuable advice.

Our responses to the reviewers’ comments are as follows:

Responses to the comments of Reviewer #1

The authors have made substantial improvements to the manuscript based on the previous review. The explanations regarding data processing, filter settings, and synchronization have been clarified, and the manuscript is now more structured and coherent. However, a few issues remain:

Comments1. lines 133-135, synchronization of EMG, IMU, and Video: The manuscript does not provide a detailed explanation of the synchronization method. It is unclear whether synchronization was achieved via hardware or software and whether external or internal synchronization was used. Clarifying this aspect would enhance the reproducibility of the study.

Response 1: As you suggested, we have added the sentence (Page 3, lines 133-136) as follows:

“The EMG sensor with IMU were connected to an EMG data collection system with the wireless each signals and video synchronization were recorded and collected using software named EMG and Motion Tools, software version 8.7.6.0 (Cometa slr, Milan, Italy).”

  1. Formatting Issue (Lines 394–399): There is an overlap of text and abbreviations in this section. Please review and revise the formatting to ensure readability.

Response 2:

①I apologize for the inconvenience. I have corrected the abbreviations and order in the text.

・electromyography (EMG) (Page 1, lines 26-27)

・tibialis anterior (Page 1, lines 27)

・gastrocnemius lateralis (Page 1, lines 27)

・inertial measurement unit (IMU) (Page 1, lines 29)

・an IMU (Page 2, lines 60)

・2.3.1. Surface EMG (Page 3, lines 101)

・the TA, GL, VL, and BF (Page 3, lines 102)

Figure 1. EMG electrode position and IMU location.        

・Surface EMG signal were recorded on the tibialis anterior (TA), gastrocnemius lateralis (GL) (Page 6, lines 241-242)

②I have added and corrected the order of abbreviations in this  "Abbreviations section" .

CC

co-contractions

EMG

electromyography

TA

tibialis anterior

GL

gastrocnemius lateralis

VL

vastus lateralis

BF

biceps femoris

IMU

inertial measurement unit

CCI

co-contraction index

LS

Locomotive syndrome

AP

ankle plantar flexor power

STV

stride time variability

BMI

Body mass index

Reviewer 2 Report

Comments and Suggestions for Authors

My questions have been addressed.

Author Response

Thank you very much for providing important comments.